# User Pairing for Delay-Limited NOMA-Based Satellite Networks with Deep Reinforcement Learning

**DOI:** 10.3390/s23167062

**Published:** 2023-08-09

**Authors:** Qianfeng Zhang, Kang An, Xiaojuan Yan, Hongxia Xi, Yuli Wang

**Affiliations:** 1Guangxi Key Laboratory of Ocean Engineering Equipment and Technology, Qinzhou 535011, China; qfzhang19@163.com (Q.Z.); xi15007778221@163.com (H.X.); wangyuli95@163.com (Y.W.); 2Key Laboratory of Beibu Gulf Offshore Engineering Equipment and Technology (Beibu Gulf University), Education Department of Guangxi Zhuang Autonomous Region, Qinzhou 535011, China; 3Sixty-Third Research Institute, National University of Defense Technology, Nanjing 210007, China; ankang@nuaa.edu.cn; 4School of Information Science and Engineering, Southeast University, Nanjing 210096, China

**Keywords:** NOMA-based satellite networks, delay QoS limitation, user pairing

## Abstract

In this paper, we investigate a user pairing problem in power domain non-orthogonal multiple access (NOMA) scheme-aided satellite networks. In the considered scenario, different satellite applications are assumed with various delay quality-of-service (QoS) requirements, and the concept of effective capacity is employed to characterize the effect of delay QoS limitations on achieved performance. Based on this, our objective was to select users to form a NOMA user pair and utilize resource efficiently. To this end, a power allocation coefficient was firstly obtained by ensuring that the achieved capacity of users with sensitive delay QoS requirements was not less than that achieved with an orthogonal multiple access (OMA) scheme. Then, considering that user selection in a delay-limited NOMA-based satellite network is intractable and non-convex, a deep reinforcement learning (DRL) algorithm was employed for dynamic user selection. Specifically, channel conditions and delay QoS requirements of users were carefully selected as state, and a DRL algorithm was used to search for the optimal user who could achieve the maximum performance with the power allocation factor, to pair with the delay QoS-sensitive user to form a NOMA user pair for each state. Simulation results are provided to demonstrate that the proposed DRL-based user selection scheme can output the optimal action in each time slot and, thus, provide superior performance than that achieved with a random selection strategy and OMA scheme.

## 1. Introduction

Due to the inherent nature of providing vast coverage and economic service, satellite communication has the ability to effectively supplement terrestrial networks during disasters and in rural and deserts areas; thus, it has been considered as an important component for next-generation wireless networks [1]. However, the dramatically increased demand for data access can result in even bigger challenges, including massive connectivity, limited power/spectral resources, and various quality of service (QoS) requirements, in future satellite networks. Recently, non-orthogonal multiple access (NOMA) schemes, including power domain NOMA [2] and code domain NOMA [3], featuring multiple access, high resource utilization efficiency, and user fairness, has become a promising solution to alleviate these challenges faced by future satellite networks. Of these two schemes, power domain NOMA (or simply NOMA for short) scheme, which has the ability to harmoniously integrate with orthogonal multiple access (OMA) techniques in existing satellite architectures, is the main motivation and focus of this article.

In a NOMA-based satellite network, a satellite/multiple users can simultaneously communicate with multiple users/a satellite in downlink/uplink transmissions by superposing various signals with different power levels in the same time/spectral block. To date, many works have investigated the performance enhancement of various NOMA-based satellite networks, such as the improved outage probability in integrated satellite terrestrial relay networks with perfect successive interference cancellation (SIC) [4], augmented erogodic capacity in uplink satellite communications [5], and increased network utility in the satellite-based internet of things [6]. An extension of work [4] to an imperfect SIC scenario with Alamouti space–time block coding was studied in [7]. Moreover, some works studied resource management of the NOMA-based satellite network from the perspective of increasing system resource efficiency, e.g., aiming to maximize long-term age of information, the authors in [8] utilized a ListNet algorithm and a particle swarm optimization algorithm to obtain an optimized power allocation solution in a satellite-based internet of things scenario. Similarly, the work in [9] proposed a joint subchannel assignment and power allocation algorithm to further optimize the sum rate of a secondary network in cognitive satellite–unmanned aerial vehicle–terrestrial networks. Although NOMA-based satellite networks can enhance spectrum/power utilization efficiency and a system’s performance, we must note that these Shannon performance enhancements were achieved by selecting users with distinctive differences in channel gains to form a NOMA pair, which is only suitable for a system with delay-insensitive applications.

However, with rising technological developments in wireless communications, new satellite applications with diverse delay QoS requirements have occurred to facilitate our daily life and provide more efficient service, such as applications on smart grids, environmental monitoring and forecasting, navigation, smart cities, and telemedicine. Among these applications, telemedicine and smart grids are identical delay-critical scenarios, while environmental monitoring is a typical delay-tolerant scenario. Thus, Shannon capacity, which fails to take users’ diverse delay QoS requirements into consideration, is no longer suitable to use in future satellite networks to characterize the performance of real-time and delay-sensitive applications/scenarios, and it is of paramount importance to study the achievable performance of satellite networks under heterogenous delay QoS requirements. Under these conditions, the concept of effective capacity, which was proposed in [10] as an effective performance metric to show the maximum constant arrival rate with a given delay QoS constraint, has been introduced in various satellite communication scenarios to show the adverse impact of delay QoS limitations on system performance [11,12,13,14], such as the authors in [11], who proposed an algorithm to schedule users in different time slots while guaranteeing users’ delay QoS requirements in a satellite–terrestrial backhaul network. In cognitive satellite–terrestrial networks, the effective capacity was introduced to guarantee the delay requirement of a primary user [12], whose extension to study effective energy efficiency of the same networks was studied in [13]. Moreover, work [14] studied the achieved effective capacity of a NOMA-based satellite system with delay adhering to users’ service requirements. Although these aforementioned works have shown the negative impact of delay QoS requirements on OMA-/NOMA-based satellite networks, how to select users in a NOMA based system, i.e., whether it is effective to only select users with big channel differences, to form a NOMA pair has not been investigated.

It is worth noting that, in addition to free space loss (FSL), antenna gain, fading severity, and location information in a beam spot can also influence the link budget of a satellite user, all of which, combined with users’ various delay QoS requirements, make the user grouping in a NOMA-based system nontrivial, especially in satellite networks, which are highly applied in military and civilian fields. To solve this challenge, a supervised learning algorithm, with which solutions can be obtained without model-oriented analysis and design, as an effective solution for resource management has been widely used in several prior works, such as work [15], which proposed a genetic algorithm (GA)-improved support vector machine scheme to effectively pair users for NOMA-based satellite networks. A fully connected deep neural network-assisted approach was studied in [16,17] to facilitate efficient beam hopping and design beam illumination pattern in multibeam satellite systems, respectively. The work in [18] proposed an accurate forecasting method by using deep neural networks for LEO satellite links. Notably, supervised learning, such as the algorithms used in [15,16,17,18], needs to learn characteristics from input data and desired output data, while a reinforcement learning (RL) algorithm, which is model-free and data-driven, has been extensively adopted in various wireless networks with different objectives. For example, based on *Q*-learning, an algorithm for jointly optimizing user pairing and power allocation was proposed in [19] to maximize the total sum rate of a satellite random access system. Considering large-scale low-earth orbit constellations, the work in [20] developed a low-complexity successive deep Q-learning algorithm for optimal satellite handover. The authors in [21] proposed a *Q*-learning NOMA-based random access scheme for time slot and channel allocation in satellite–terrestrial relay networks. In [22], the authors adopted a graph neural network and RL algorithms in a hybrid satellite–terrestrial network to optimize UAV trajectory and maximize the number of served users. In [23,24], the authors conducted resource management in a relay-aided network with the help of distributionally robust deep RL (DRL) and enhanced DRL algorithms, respectively.

Motivated by these observations, for the work herein, we leaned upon a DRL algorithm to pair users and provide services with various delay QoS requirements for future NOMA-based satellite networks (since this paper’s aim was to pair users in delay-limited NOMA-based satellite networks with a DRL algorithm, while the impacts of low-density parity check codes [25] in NOMA-based satellite networks will be our follow-up research.). The main contributions of this work can be described as follows:The concept of effective capacity is employed to measure the rate achieved with a given delay QoS constraint, based on which, a power allocation coefficient is firstly obtained by ensuring the achieved capacity of users with sensitive delay QoS requirements is not less than that achieved with an OMA scheme, and then, the user pairing problem is formulated with the aim of maximizing the sum effective capacity of the considered system;Because various delay QoS requirements have varying negative impacts on users’ capacity, user pairing in a NOMA-based network with various delay QoS constraints is different from that in traditional NOMA-based delay-insensitive system. In this condition, to maximize system capacity with the obtained power allocation factor, when the delay-critical user is fixed, a DRL approach is introduced to select one user who has relatively insensitive delay requirement and good link condition, compared to the other users, to optimize NOMA user pairing with low complexity;The proposed DRL-based NOMA user pairing strategy is compared to an OMA scheme and NOMA with a random user-selecting scheme, which reveal the superiority of introducing the NOMA scheme and DRL algorithm in the satellite networks from the perspective of performance enhancement. Specifically, the advantage of the proposed approach is achieved by selecting the most suitable delay tolerant user to pair with the delay-sensitive user and form a NOMA user group in each time slot.

The rest of this paper is outlined as follows. The system model is presented in Section 2. Section 3 introduces the concept of effective capacity, obtains the power allocation scheme by ensuring the achieved capacity of the user with sensitive delay QoS requirement is not less than that achieved with the OMA scheme, and formulates the user pairing problem for the delay-limited NOMA-aided satellite network. In Section 4, a DRL algorithm is described in detail and tested in the proposed system. Performance results are discussed and conclusions are given in Section 5 and Section 6, respectively.

## 2. System Model

Consider a downlink NOMA-based satellite system that is designed to serve *m* (m≥2) users with the help of the NOMA scheme. These *m* users are randomly deployed in an area approximated as a circle of radius *R* with different channel statistical prosperities and delay QoS requirements. (In this paper, channel estimation errors, co-channel interference, complexity, and mobility constraints are not taken into consideration in the proposed system model; the influences of these parameters on user selection and system performance will be a focus in our future works, based on the contributions in the current work.) Without loss of generality, users are ordered based on their link budgets, i.e., $Q_{1}$ ≤ $Q_{2}$⋯ ≤ $Q_{m}$, where $Q_{j}$ is the link budget of User *j* (*j* = 1, 2, ⋯, *m*). For simplicity, we further assume only the $c^{th}$ and $t^{th}$ users (1 ≤ *c* < *t* ≤ *m*) are selected to form a NOMA group, and each user in the proposed model is equipped with a single antenna.

Thus, the received signal at User *j*(j=c,t) is
(1)$$y_{j}=Q_{j}x+w_{j},$$
where $w_{j}$ denotes the noise at User *j* with zero mean and $\delta^{2}$ variance, $x=\sum_{j=c,t}\sqrt{\alpha_{j}^{p}P_{s}}x_{j}$ is the superposed signal (with $\alpha_{j}^{p}$ being a fraction of the transmission power $P_{s}$ allocated to User *j* and $x_{j}$$\left(E[\right|x_{j}{|}^{2}]=1)$ being the signal for User *j*), $Q_{j}$ (including FSL, antenna gain, beam gain, and fading model) is the entire link budget from satellite to User *j*, which can be described as follows:(2)$$Q_{j}=\Phi_{j}G_{s}\left(\varphi_{j}\right){\left|g_{j}\right|}^{2},$$
where $\Phi_{j}=L_{j}G_{j}$, with $L_{j}$ and $G_{j}$ being the FSL and antenna gain at User *j*, respectively. $G_{s}\left(\varphi_{j}\right)$, which is the beam gain of User *j*, with $\varphi_{j}$ denoting the angle between User *j* and beam center with respect to the satellite, can be approximated as [5]
(3)$$G_{s}\left(\;\varphi_{j}\;\right)\;\approx \;G_{\max}\;{\left(\;\frac{J_{1}\left(\;ad_{j}\;\right)}{2ad_{j}}\;+\;36\frac{J_{3}\left(\;ad_{j}\;\right)}{a^{3}d_{j}^{3}}\right)}^{2}\;\;\;=\;G_{s}\left(\;d_{j}\;\right),$$
with $G_{\max}$ representing the maximum antenna gain, $J_{n}\left(\cdot \right)$ being the Bessel function of first kind and *n*-th order, $d_{j}$ being the distance from the beam center to User *j*, and a=2.07123/R. ${\left|g_{j}\right|}^{2}$ is the channel power gain of the satellite link, which is assumed to follow a widely applied Shadowed Rician fading model [26,27,28,29,30]. According to [31], the probability density function (PDF) of ${\left|g_{j}\right|}^{2}$ is
(4)$$f_{{\left|g_{j}\right|}^{2}}\left(x\right)=\alpha_{j}e^{-\beta_{j}x}{}_{1}F_{1}\left(m_{j};1;\delta_{I}x\right),$$
where $\alpha_{j}=\frac{{\left(2b_{j}m_{j}\right)}^{m_{j}}}{2b_{j}{\left(2b_{j}m_{j}+\Omega_{j}\right)}^{m_{j}}}$, $\delta_{j}=\frac{\Omega_{j}}{2b_{j}\left(2b_{j}m_{j}+\Omega_{j}\right)}$, $\beta_{j}=\frac{1}{2b_{j}}$ with $2b_{j}$ and $\Omega_{j}$, respectively, being the average power of the multipath and the LoS components, $m_{j}$$\left(m_{j}>0\right)$ denoting the Nakagami-*m* fading parameter, and ${}_{1}F_{1}\left(a;b;c\right)$ representing the confluent hypergeometric function ([32], Equation (9.14.1)).

Based on the principle of the downlink NOMA scheme, decoding order is decided by users’ channel qualities, i.e., the user with a worse link condition decodes its own information firstly and directly. Thus, the signal-to-interference-plus-noise ratio (SINR) of User *c* is
(5)$$\gamma_{c}^{N}=\frac{\alpha_{c}^{p}\gamma \Phi_{c}G_{s}\left(d_{c}\right){\left|g_{c}\right|}^{2}}{\alpha_{t}^{p}\gamma \Phi_{c}G_{s}\left(d_{c}\right){\left|g_{c}\right|}^{2}+1^{}}=\frac{\alpha_{c}^{p}\gamma Q_{c}}{\alpha_{t}^{p}\gamma Q_{c}+1},$$
where $\alpha_{c}^{p}\;+\;\alpha_{t}^{p}\;=\;1$ and $\gamma \;=\;P_{s}/\delta_{}^{2}$ is the average transmission SNR. At the same time, the user with better channel quality, i.e., User *t*, adopts the SIC strategy to decode and remove the interference from User *c*; the decoding SINR can be derived as
(6)$$\gamma_{t\to c}^{N}=\frac{\alpha_{c}^{p}\gamma \Phi_{t}G_{s}\left(d_{t}\right){\left|g_{t}\right|}^{2}}{\alpha_{t}^{p}\gamma \Phi_{t}G_{s}\left(d_{t}\right){\left|g_{t}\right|}^{2}+1^{}}=\frac{\alpha_{c}^{p}\gamma Q_{t}}{\alpha_{t}^{p}\gamma Q_{t}+1}.$$

We can derive that $\gamma_{c}^{N}\;<\;\gamma_{t\to c}^{N}$, since $Q_{c}\;<\;Q_{t}$. Then, User *t* decodes its own information, and the achieved SINR is
(7)$$\gamma_{t}^{N}=\alpha_{t}^{p}\gamma \Phi_{t}G_{s}\left(d_{t}\right){\left|g_{t}\right|}^{2}\;=\alpha_{t}^{p}\gamma Q_{t}.$$

## 3. Effective Capacity and Power Allocation

### 3.1. Effective Capacity

To provide services with different delay QoS requirements, the concept of effective capacity is employed to characterize the effect of delay QoS limitation on achieved performance, characterized by θ(θ≥0) [10]. In this paper, the uncorrelated service process across different slots is further assumed and the normalized effective capacity is adopted. Under these conditions, given a delay QoS exponent $\theta_{j}$, the normalized effective capacity of User *j* in bps/Hz is
(8)$$\begin{array}{c} C_{j}\left(\theta_{j}\right)=\frac{-1}{\theta_{j}T_{f}B}\ln\left(E\left\{e^{-\theta_{j}T_{f}B\;R_{j}}\right\}\right)=\frac{1}{\psi_{j}\ln2}\ln\left(E\left[{\left(1+\gamma_{j}\right)}^{\psi_{j}}\right]\right), \end{array}$$
where $\psi_{j}\;=\;-\theta_{j}T_{f}B/\ln2$, with $T_{f}$ and *B* being the frame duration and the occupied bandwidth, respectively, $R_{j}=\log_{2}\left(1+\gamma_{j}\right)$ is User *j*’s transmission rate, and E is the expectation operator. We note that a larger/smaller delay QoS exponent $\theta_{j}$ is required in a more critical/tolerant delay-limited scenario.

### 3.2. Power Allocation Strategy

To ensure the capacity achieved by the user with a critical delay QoS requirement using the NOMA scheme is always better than that with the TDMA scheme, the power allocation coefficient should be further constrained. In this section, a power allocation scheme is investigated for two cases, i.e., User *c* in Case 1 and User *t* in Case 2 are assumed to be delay-sensitive users.

For Case 1, $\theta_{c}>\theta_{t}$ is assumed and the power allocation factor is limited by $C_{c}^{N}\left(\theta_{c}\right)\geq C_{c}^{T}\left(\theta_{c}\right)$, where
(9)$$\begin{array}{c} C_{c}^{N}\left(\theta_{c}\right)=\frac{1}{\psi_{c}\ln2}\ln\left(E\left[{\left(1+\gamma_{c}^{N}\right)}^{\psi_{c}}\right]\right), \end{array}$$
and
(10)$$\begin{array}{c} C_{c}^{T}\left(\theta_{c}\right)=\frac{1}{\psi_{c}\ln2}\ln\left(E\left[{\left(1+\gamma_{c}^{T}\right)}^{0.5\psi_{c}}\right]\right), \end{array}$$
with $\gamma_{j}^{T}=\gamma \Phi_{j}G_{s}\left(d_{j}\right){\left|g_{j}\right|}^{2}=\gamma Q_{j}$ being the SINR of User *j* (j=c,t) achieved with the TDMA scheme, and 0.5 owes to the loss in multiplexing in the TDMA system. By substituting (5) into (8), along with some manipulations, $\alpha_{c}^{p}$ can be derived as
(11)$$\begin{array}{c} \alpha_{c}^{p}\geq 1-\frac{1}{\sqrt{1+\gamma Q_{c}}+1}, \end{array}$$
which means that the value of $\alpha_{c}^{p}$ is decided by γ, location information, and fading severity of User *c*.

For Case 2, $\theta_{t}>\theta_{c}$ is considered, and factor $\alpha_{c}^{p}$ is limitied by restriction condition $C_{t}^{N}\left(\theta_{t}\right)\geq C_{t}^{T}\left(\theta_{t}\right)$, with
(12)$$\begin{array}{c} C_{t}^{N}\left(\theta_{t}\right)=\frac{1}{\psi_{t}\ln2}\ln\left(E\left[{\left(1+\left(1-\alpha_{c}^{p}\right)\gamma Q_{t}\right)}^{\psi_{t}}\right]\right), \end{array}$$
and
(13)$$\begin{array}{c} C_{t}^{T}\left(\theta_{t}\right)=\frac{1}{\psi_{t}\ln2}\ln\left(\;E\left[{\left(1+\gamma Q_{t}\right)}^{0.5\psi_{t}}\right]\right). \end{array}$$

Then, we can obtain
(14)$$\begin{array}{c} \alpha_{c}^{p}\leq 1-\frac{1}{\sqrt{1+\gamma Q_{t}}+1}. \end{array}$$

Based on the power allocation coefficient obtained in (11) for Case 1 or (14) for Case 2, the effective capacity of User *c* can be given by
(15)$$\begin{array}{ccc} C_{c}^{N}\left(\theta_{c}\right) & = & \frac{1}{\psi_{c}\ln2}\ln\left(E\left[{\left(1+\gamma_{c}^{N}\right)}^{\psi_{c}}\right]\right)\\ & = & \frac{1}{\psi_{c}\ln2}\ln\left(\int_{R_{jn}}^{R_{jf}}\int_{0}^{\infty }{\left(\;1+\gamma_{c}^{N}\right)}^{\psi_{c}}f_{{\left|g_{c}\right|}^{2}}^{}\left(x\right)f_{d_{c}}\left(y\right)dxdy\right), \end{array}$$
where $f_{d_{j}}\;\;\left(y\right)\;=\;\frac{2y}{R_{jf}^{2}-R_{jn}^{2}}$ is the PDF of User *j*’s location [4] if it distributes in an annular area with inner radius $R_{jn}$ and outer radius $R_{jf}$. To evaluate (15), we first express ${}_{1}F_{1}\left(m_{j};1;\delta_{j}x\right)$ in (4) and ${\left(1+x\right)}^{a\;}$ in terms of the Meijer G-functions from Equation (9.34.8) in [32] and binominals represented by Equation (1.11) in [32], as 
(16)$$\begin{array}{c} {}_{1}F_{1}\left(m_{j};1;\delta_{j}x\right)=\frac{1}{\Gamma \left(m_{j}\right)}G_{1,2}^{1,1}\left[-\delta_{j}x\left|\begin{array}{c} 1-m_{j}\\ 0,0 \end{array}\right.\right], \end{array}$$
and
(17)$$\begin{array}{c} {\left(1+x\;\;\right)}^{a\;}=\sum_{k=0}^{\infty }\frac{\Gamma \left(a+k\right)}{k!\Gamma \left(a\right)}{}^{}\;x^{k}, \end{array}$$
where $G_{1,2}^{1,1}\left[\cdot |\cdot \right]$ ([32], Equation (9.301)) is the Meijer-G function and $\Gamma \left(\cdot \right)$ ([32], Equation (8.310.1)) is the Gamma function. Then, inserting (4), (5), (16)–(17) into (15) along with ([32], Equation (7.813.1)), we obtain the result as
(18)$$\begin{array}{cc} C_{c}^{N}\left(\theta_{c}\right) & =\frac{1}{\varphi_{c}\ln2}\ln\left(\alpha_{c}\sum_{k=0}^{\infty }\sum_{m=0}^{\infty }\frac{{\left(1-\alpha_{c}^{p}\right)}^{k}\Phi_{c}^{m+k}\gamma^{m+k}G_{\max}\Gamma \left(m+\varphi_{c}\right)\Gamma \left(k-\varphi_{c}\right)}{\beta_{c}^{k+m+1}k!m!\Gamma \left(m_{c}\right)\Gamma \left(-\varphi_{c}\right)\Gamma \left(\varphi_{c}\right)}\right.\\ & \times G_{2,2}^{1,2}\left[-\frac{\delta_{c}}{\beta_{c}}\left|\begin{array}{c} -k-m,1-m_{c}\\ 0,0 \end{array}\right.\right]\left.\int_{R_{cn}}^{R_{cf}}{\left(\frac{J_{1}\left(ay\right)}{2ay}+\frac{J_{3}\left(ay\right)}{a^{3}y_{}^{3}}\right)}^{2}\frac{2y}{R_{cf}^{2}-R_{cn}^{2}}dy\right). \end{array}$$

By further defining $\Psi_{c}$ to denote the integration part of (18) and, with the help of Equation (8.442.2) in [32], we obtain
(19)$$\begin{array}{c} \Psi_{c}=\sum_{n=0}^{\infty }\Theta \left(n\right)\frac{{\left(-1\right)}^{n}a^{2n}\left(R_{cf}^{2n+2}-R_{cn}^{2n+2}\right)}{n!4^{n}\left(2n+2\right)\left(R_{cf}^{2}-R_{cn}^{2}\right)}, \end{array}$$
where
(20)$$\begin{array}{c} \Theta \left(n\right)=\frac{F\left(-n,-1-n;2;1\right)}{16\Gamma \left(2+n\right)}+\frac{F\left(-n,-1-n;4;1\right)}{8\Gamma \left(4\right)\Gamma \left(2+n\right)}+\frac{F\left(-n,-3-n;4;1\right)}{32\Gamma \left(4\right)\Gamma \left(4+n\right)}, \end{array}$$
with $F\left(a,b;c;d\right)$ being the hypergeometric function ([32], Equation (9.100)). Finally, substituting (19) and (20) into (18), the desired result for the expression of $C_{c}^{N}\left(\theta_{c}\right)$ can be obtained.

Similarly, the effective capacity of User *t* can be given by
(21)$$\begin{array}{c} C_{t}^{N}\left(\theta_{c}\right)=\frac{1}{\psi_{t}\ln2}\ln\left(\int_{R_{tn}}^{R_{tf}}\int_{0}^{\infty }{\left(\;1+\gamma_{t}^{N}\right)}^{\psi_{t}}f_{{\left|g_{t}\right|}^{2}}^{}\left(x\right)f_{d_{t}}\left(y\right)dxdy\right). \end{array}$$

By substituting (4) and (7) into (21) and following with the similar steps as those in the derivation of (12), the effective capacity expression of User *t* can be derived as
(22)$$\begin{array}{cc} C_{t}^{N}\left(\theta_{t}\right)= & \frac{1}{\varphi_{t}\ln2}\ln\left[\sum_{k=0}^{\infty }\frac{\alpha_{t}\Gamma \left(\varphi_{t}+k\right){\left(\alpha_{t}^{p}\bar{\gamma }\Phi_{t}\right)}^{k}}{k!\Gamma \left(\varphi_{t}\right)\Gamma \left(m_{t}\right)\beta_{t}^{k+1}}G_{2,2}^{1,2}\left(\frac{-\delta_{t}}{\beta_{t}^{}}\left|\begin{array}{c} -k,1-m_{t}\\ 0,0 \end{array}\right.\right)\right.\\ & \times \left.\sum_{n=0}^{\infty }\Theta \left(n\right)\frac{{\left(-1\right)}^{n}a^{2n}\left(R_{cf}^{2n+2}-R_{cn}^{2n+2}\right)}{n!4^{n}\left(2n+2\right)\left(R_{cf}^{2}-R_{cn}^{2}\right)}\right]. \end{array}$$

Then, the sum effective capacity of the considered system can be given as $C^{N}=C_{c}^{N}\left(\theta_{c}\right)+C_{t}^{N}\left(\theta_{t}\right)$.

### 3.3. Problem Formulation

Although the closed-form expression of sum rate for the considered system has been derived, we must note that the rate of User *j* (j=c,t) is influenced by many factors, such as delay exponent $\theta_{j}$, transmission SNR γ, fading severity, location information $d_{j}$, and $\alpha_{j}^{p}$. Thus, to expressively show the different impacts of these key parameters on the achieved performance, the normalized effective capacity of User *j* is plotted in Figure 1, where ILS, AS, and FHS are infrequent light shadowing, average shadowing, and frequent heavy shadowing, respectively.

From Figure 1, we can directly observe that, when $\theta_{j}\to 0^{}$, effective capacity converges to the ergodic capacity, since only delay-insensitive traffic is needed. However, when $\theta_{j}>10$, even for case $\alpha_{j}^{p}=1$, effective capacity reduces to 0 due to the required delay QoS being too stringent. Thus, the range of User *j*’s delay limitation is assumed to be constrained as $\theta_{j}\in \left[0.5,10\right]$ in this paper. In addition, an increased $d_{j}$, i.e., a worse fading severity, or decreased γ can degrade the capacity curves. Moreover, all capacity curves decrease with increasing $\theta_{j}$. This observation clearly indicates that the achieved performance suffers from a combination of these factors, although, in both Case 1 and Case 2, it seems like a user with the smallest delay QoS exponent, nearest location information, and best shadowing should be selected as User *t*/*c* and paired with the User *c*/*t* in Case 1/2 to maximize the sum performance of the considered system. Conversely, while in a spot beam, the user with the nearest location information or best fading condition may have a relatively large θ, or vice versa. Thus, how to select User *t*/*c* in Case 1/2 is a vital issue in a delay-limited scenario.

For simplicity, herein, we mainly focus on the user pairing in Case 1, which means that $\alpha_{c}^{p}$ must meet $C_{t}^{N}\left(\theta_{t}\right)\geq C_{t}^{T}\left(\theta_{t}\right)$. Then, the optimization problem is to find a user who can obtain the best power utilization efficiency, after taking into account link budget and delay QoS requirement, to be the User *t*. The mathematical formulation of this problem can be denoted by P1 and formulated as
(23)$$\begin{array}{cc} P1 & :\max_{\left\{d_{t},Q_{t},\theta_{t}\right\}}\;C_{t}^{N}\left(\theta_{t}\right)\\ s.t.\;\;\;C1 & :Q_{t}>Q_{c},t\in 1,2,\cdots ,c-1,c+1,\cdots ,m;\\ C2 & :\left(11\right),\theta_{c}>\theta_{t}>0;\\ C3 & :d_{c,t}\leq R. \end{array}$$

In the aforementioned problem, C1 ensures that the link budget of User *t* must be better than that of User *c* to successfully perform SIC; C2 denotes that, in Case 1, the resource allocation threshold in (11) must be ensured to guarantee the minimum data rate requirement of User *c*, and C3 implies that the limited location information of Users *c* and *t*.

## 4. DRL for Delay-Constrained User Pairing

The deep *Q*-network (DQN) algorithm, which combines the advantages of *Q*-learning and deep neural networks, is one of the most representative value-based method in the DRL family, with which the expected returns of actions can be predicted based on a certain environmental observation; a framework of applying such an algorithm in user pairing for the considered system is provided in Figure 2. (The DQN method is the classical approach in the DRL family, whose complexity analysis is not provided in this paper—the interested reader can refer to [33].) Although DRL deployment causes more delay, it is believed that this delay can be significantly decreased with the improvement of chip processing speed.).

Since our objective in problem P1 is to choose an appropriate user to be User *t* at different time slots to maximize the power resource utilization, we thus define a tuple $\bar{M}:=<S,A,R,\pi >$ to model this problem as a Markov decision process (MDP) for a stationary decision. Specifically, *S* means the state and observation space, *A* represents the set of actions, *R* means the designed reward, and π is the policy that makes the decision. Meanwhile, $Q_{\pi }\left(s_{l},a_{l}\right)$ is defined as the Q-value obtained with policy π when the environment is in state $s_{l}$ while adopting action $a_{l}$ at the $l^{th}$ time slot. For the problem P1, key elements, such as the states, actions, and cost, in an MDP model are described in detail as follows:State *S*: At time slot *l*, a tuple denoted by $s_{l}=\left(P_{s},\Phi_{j},d_{j},g_{j},\theta_{j}\right)$, $s_{l}\in S$ is used to describe the system state, where $P_{s},\Phi_{j},d_{j},g_{j},\theta_{j}$ are transmission power, antenna gains, location information, fading severity, and delay QoS exponent of User *j*(j=1,2,⋯,c−1,c+1,⋯,m), as analyzed in Section 2 and Section 3, respectively. Since $s_{l}$ varies in different time slots, the agent is required to adjust its action in each slot accordingly;Action *A*: NOMA user pairing is important for NOMA-aided satellite networks with delay QoS constraints because it directly impacts the resource utilization efficiency. Thus, user selection should be designed based on current state; here, we set the action space as [A=1,2,⋯,c−1,c+1,⋯,m], and then $a_{l}=m$ means the $m^{th}$ user is selected to be the User *t*;Reward design: Equation (11) must be satisfied to ensure that User *c*’s performance achieved with the NOMA scheme is not less than that achieved with the TDMA scheme. Based on this, our objective is to select a user to be User *t* who, with the remaining power resource, can achieve the largest effective capacity. Thus, if User *j* is selected at time slot *l*, the reward is assigned as
(24)$$\begin{array}{c} C_{j}^{N}\left(\theta_{j},l\right)=C_{j}^{N}\left(\theta_{j}\right),\left(j=1,2,\cdots ,c-1,c+1,\cdots ,m\right) \end{array}$$

As can be seen from Figure 2, the DQN algorithm has two phases. In the data-generation phase, *Q*-learning with experience pool *D* is used to generate data for the next network-training phase. In this process, the agent chooses an action $a_{l}$ according to its observation $s_{l}$ under policy π. To trade off between exploration and exploitation, ε-greedy exploration is used here, which means, for state $s_{l}$, a random action with probability ϵ (0<ϵ<1) or the best action with probability (1−ϵ) is chosen to be action $a_{l}$. With this ε-greedy policy, *Q*-value function $Q_{\pi }\left(s_{l},a_{l}\right)$, which describes the expected $R_{\pi }\left(l_{}\right)$, can be given by
(25)$$\begin{array}{c} Q_{\pi }\left(s_{l},a_{l}\right)=E\left(R_{\pi }\left(l\right)\left|S=s_{l},A=a_{l}\right.\right). \end{array}$$

This *Q*-value function is updated with
(26)$$Q_{\pi }\left(s_{l},a_{l}\right)=Q_{\pi }\left(s_{l},a_{l}\right)\left(1-\bar{\alpha }\right)+\bar{\alpha }\left(R_{\pi }\left(l\right)+\underset{a_{l+1}}{\hat{\gamma }\max}\;Q_{\pi }\left(s_{l+1},a_{l+1}\;\right)\right),$$
where $\bar{\alpha }$ and $\hat{\gamma }$ are the learning rate and discount factor, respectively. The best action can be written as $a_{l}^{★}=\underset{a_{l}\in A}{max}Q_{\pi }\left(s_{l},a_{l}\right)$. Following the environmental transition resultant from variations in users’ link budgets and delay QoS limitations, the tuple $\left(s_{l},a_{l},R_{l},s_{l+1}\right)$ at the $l^{th}$ time slot is collected and stored in the experience pool, in which the old tuple gives space to the newest tuple (if the pool is full).

Considering that the number of satellite users in a beam spot could be very large, the size and computation efficiency of *Q* values of (25) for all possible actions are large and low. In this context, deep neural networks parameterized by $\theta^{{}^{\prime }}$ and θ, called target DQN and training DQN, respectively, are used in the neural network training phase to estimate the *Q*-value by function approximations. As shown in Figure 2, the target of the DQN is to estimate the maximum *Q*-value for the next state, i.e., $\max_{a_{t+1}}Q\left(s_{t+1},a_{t+1};\theta^{{}^{\prime }}\right)$. The training DQN network is deployed to make an action decision and estimate the *Q*-value for the current state, whose loss function can be written as
(27)$$L\left(\theta \right)=E\left[{\left(R_{\pi }\left(t\right)+\max_{a_{t+1}^{}}Q\left(s_{t+1},a_{t+1};\theta^{{}^{\prime }}\right)-Q\left(s_{t},a_{t};\theta \right)\right)}^{2}\right].$$

Using stochastic gradient descent to minimize the function in (27), the correct weights of $\theta^{}$ can be learned by the training DQN. The weights $\theta^{{}^{\prime }}$ are frozen for several steps and then updated by setting $\theta^{{}^{\prime }}=\theta^{}$ for the goal of stabilizing the training. The specific steps for the training DQN to select one from many users to be User *t* is given in Algorithm 1.
**Algorithm 1:** DQN Algorithm-based NOMA User Pairing in Satellite Networks.
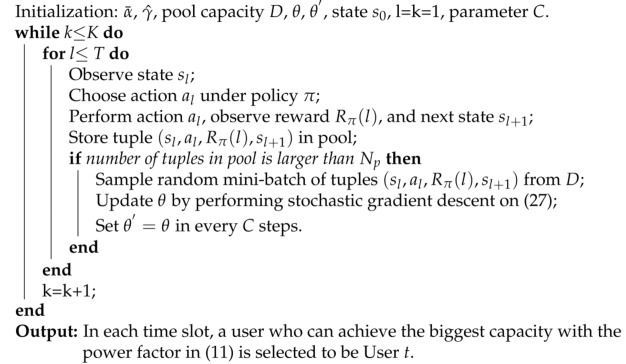



## 5. Results

In this section, simulation results are provided to characterize the effects of users’ specific delay QoS requirements on the power allocation scheme, user selection strategy, and system performance. Without loss of generality, we assumed $T_{f}B=1$, the carrier frequency as 4 GHz, and radius R=125 km [5,13]. Moreover, we set the number of users as 150, the fading severities, location information, and delay requirements of these users were randomly generated within [ILS, AS, FHS], [0, 1R], and [0.5, 10], respectively, to show the various channel conditions, locations, and application scenarios of different satellite users. The delay QoS exponent of the delay-sensitive user, i.e., User *c*, was set as $\theta_{c}=9.38$, and the label (ILS/AS) denotes the link-shadowing severity of User *t*/User *c* in this paper.

We first conducted numerical simulations to show the impact of shadowing, γ, and $d_{c}$ on the power allocation coefficient $\alpha_{c}^{p}$, as illustrated in Figure 3. From this figure, we can clearly see that, when User *c* experiences a lighter shadowing, a higher γ, or a closer location information $d_{c}$, a larger $\alpha_{c}^{p}$ is needed to ensure that the performance achieved with the NOMA scheme is not less than that achieved with the TDMA scheme, which is consistent with the analytical result given in (11). In the following simulations, $\alpha_{c}^{p}$ was set to meet the condition of $C_{c}^{N}\left(\theta_{c}\right)=C_{c}^{T}\left(\theta_{c}\right)$ without other descriptions. Moreover, it can be observed that the analytical results were all consistent with the Monte Carlo simulations.

Then, simulations were conducted to illustrate the capacity of User *t* achieved with the NOMA scheme and TDMA scheme versus delay requirement $\theta_{t}$, shown in Figure 4), from which we can clearly observe that the capacity curves all degrade with increasing $\theta_{t}$. This is an expected result because a larger $\theta_{t}$ means a smaller tolerated delay outage and a lower supported constant arrival rate. Moreover, we find that the superiority of the NOMA scheme gradually decreases with increasing delay limitation $\theta_{t}$, i.e., when $\theta_{t}\geq 10^{0.4}$, the capacity gap between NOMA and TDMA curves almost disappears. The superiority of the NOMA scheme, for the case $\theta_{t}<10^{0.4}$, is significantly upgraded for a larger γ, a lighter fading severity of User *t*, or a smaller $d_{t}$. This is because any of these factors corresponds to a more favourable condition. This phenomenon suggests that, in addition to the shadowing, $d_{t}$, and γ, $\theta_{t}$ must be taken into account to form a flexible NOMA user group and ensure the superiority of NOMA-based satellite networks.

Finally, the DQN algorithm was adopted to select one from many users to be User *t* and pair them with User *c* to form a NOMA user group. Specially, since the assumption that $Q_{c}<Q_{t}$ must be satisfied, only users with ILS/AS severity were viewed as candidates. Meanwhile, $\alpha_{t}^{p}=1-\alpha_{c}^{p}$ varied with the location and fading severity of User *c* as well as the transmission average SNR γ, as shown in Figure 3.

The convergences of the proposed DQN algorithm with different learning rates are shown in Figure 5, from which we find that a smaller value of learning rate leads to a faster convergence, since a smaller learning rate means a lower newly acquired cost is accepted to adjust the evaluated $Q_{\pi }\left(s_{l},a_{l}\right)$. Thus, $\bar{\alpha }=0.01$ was set in our algorithm. Figure 6 compares the effective capacity of selected user achieved with NOMA and TDMA schemes under the proposed strategy and random selection strategy. It can be seen from Figure 6 that curves with the proposed NOMA scheme are superior to those with the TDMA scheme for all cases, demonstrating the advantages of employing the NOMA scheme in delay QoS-constrained satellite communication networks. Moreover, since the proposed DQN-based user selection scheme can find the optimal action for each state, and, thus, it can provide superior performance as well as a much bigger performance difference between NOMA and TDMA schemes than those achieved with a random selection strategy in each time slot.

## 6. Conclusions

In this paper, we have proposed a user pairing scheme in NOMA-based satellite networks with delay QoS constraints. With the objective of maximizing the sum effective capacity without degrading the performance of the delay-sensitive user, the user pairing problem was formulated. In particular, we designed the power allocation strategy to make sure that the performance of the delay-sensitive user achieved with the NOMA scheme was not less than that achieved with an OMA scheme. Based on this, the DRL algorithm was adopted to select a user from many users to pair with the delay-sensitive user and form a NOMA group. Simulation results have been provided to validate those performance analyses, show the effects of key parameters on system performance and the user selection strategy, and demonstrate that the DRL algorithm can significantly improve the system performance by finding the optimal action for each state.

## Figures and Tables

**Figure 1 sensors-23-07062-f001:**
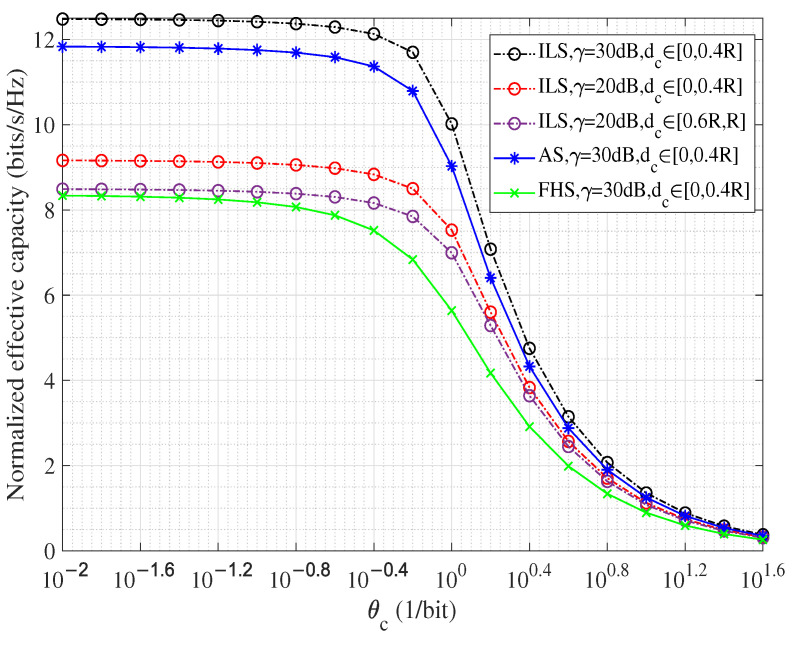
Normalized effective capacity versus delay exponent $\theta_{j}$ for various SNR γ, fading severity, and location information $d_{j}$, when $\alpha_{j}^{p}=1$.

**Figure 2 sensors-23-07062-f002:**
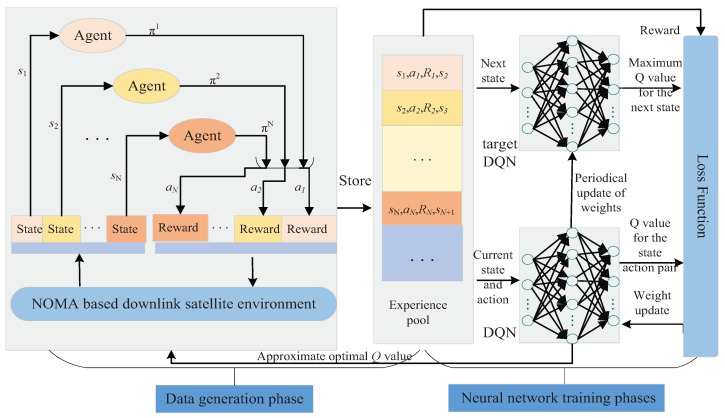
DQN-based NOMA user pairing model.

**Figure 3 sensors-23-07062-f003:**
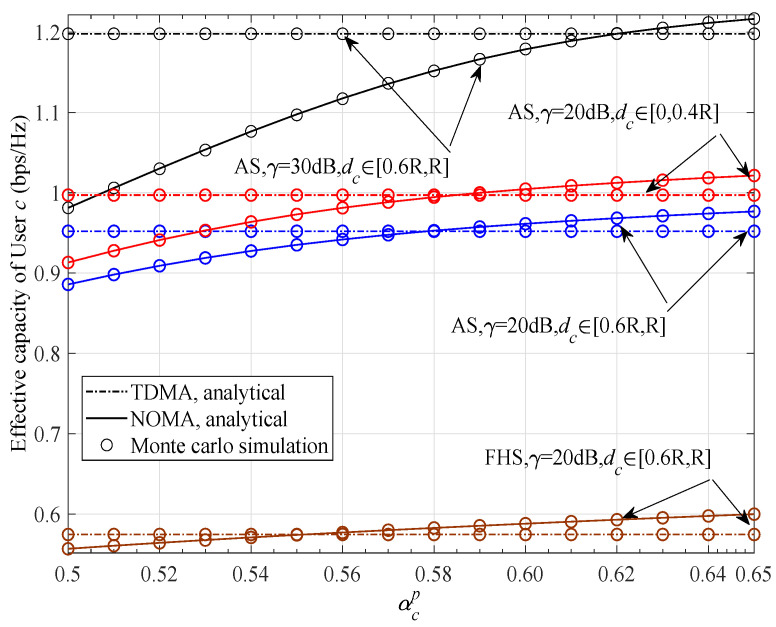
Effective capacity of User *c* achieved with TDMA and NOMA schemes versus $\alpha_{c}^{p}$ under various system parameters.

**Figure 4 sensors-23-07062-f004:**
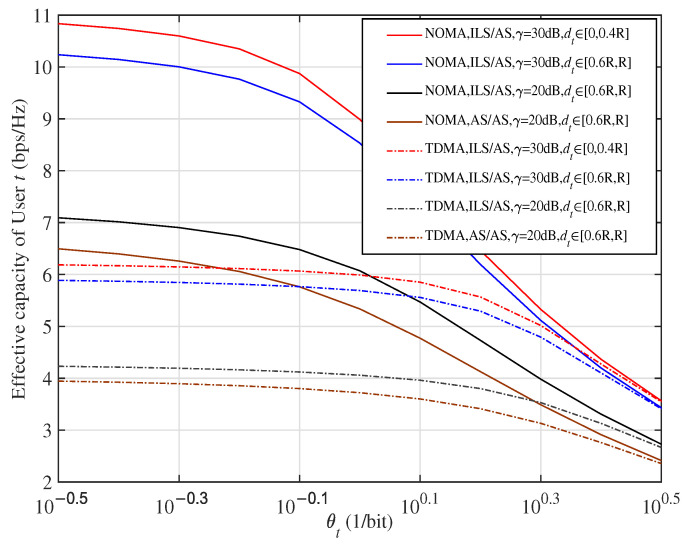
Effective capacity of User *t* for two access schemes versus $\theta_{t}$ with various γ, $d_{t}$, and fading severities, when $d_{c}\in \left[0.6R,R\right]$ and $\alpha_{t}^{p}=1-\alpha_{c}^{p}$.

**Figure 5 sensors-23-07062-f005:**
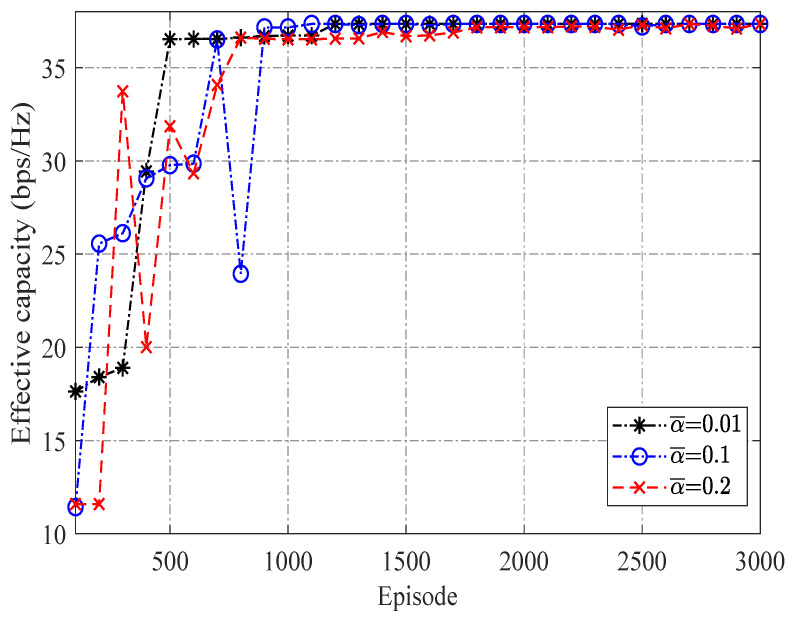
Convergences of the proposed DQN user selection algorithm with different learning rates.

**Figure 6 sensors-23-07062-f006:**
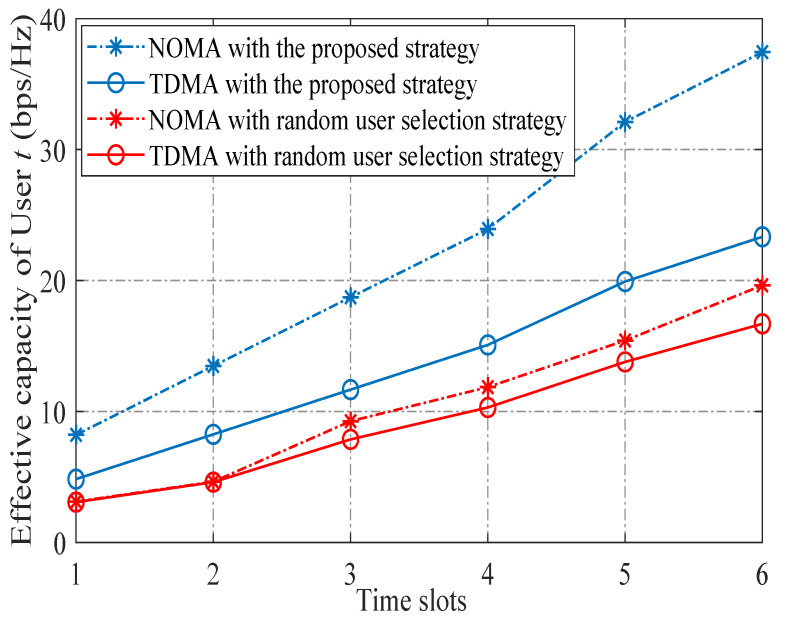
Effective capacity of selected user achieved with two access schemes under the proposed strategy and random selection strategy.

## Data Availability

Not applicable.

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
