# Peer review of "User Pairing for Delay-Limited NOMA-Based Satellite Networks with Deep Reinforcement Learning"

_sensors, 2023, doi:10.3390/s23167062_

Round 1
Reviewer 1 Report
The paper titled User Pairing for Delay Limited NOMA-Based Satellite networks with Deep Reinforcement Learning addresses the user pairing problem in satellite networks using the power domain non-orthogonal multiple access (NOMA) scheme. The authors propose a user selection scheme based on deep reinforcement learning to form NOMA user pairs and efficiently utilize network resources. Simulation results demonstrate that the proposed scheme outperforms random selection strategies and orthogonal multiple access (OMA) schemes in terms of system performance. The paper is well-written and provides a clear overview of the research conducted. However, some major issues need to be carefully addressed.
1. This manuscript only uses the effective capacity to measure the quality of the proposed system,please give a reasonable explanation.
2. The paper says that for different satellite application scenarios in the Abstract, but the corresponding conclusion is not found in the simulation part.
3. Whether there are other existing artificial intelligence algorithms used to solve this problem, the article is not clear, if so, I hope to make a comparison in the simulation.
4. In the simulation results and performance analysis, the manuscript do not consider the complexity. Besides, whether the deep reinforcement learning method used in the article is different from that in other references is not clarified. Please make some instructions.
5. The power allocation coefficient is mentioned several times in the article, but the function of this parameter is not seen in the simulation results. Please explain the reason.
6. Some state-of-the-art regarding the code-domain NOMA scheme is ignored, such using LDPC codes [A-C].
[A] “Design of capacity-approaching protograph-based LDPC coding systems,” Ph.D. dissertation, 2012.
[B] “Design of distributed protograph LDPC codes for multi-relay coded-cooperative networks,” IEEE Transactions on Wireless Communications, 2017.
[C] “Survey of turbo, LDPC, and polar decoder ASIC implementations,” IEEE Commun. Surveys Tuts., vol. 21, no. 3, pp. 2309–2333, 3rd Quart. 2019.
Please refer to the above comments.
Author Response
Dear reviewer:
Attached please find our point-to-point response. Hope you find that our response and the revision are satisfactory and the paper is suitable for its publication.
BR
Xiaojuan Yan

Reviewer 2 Report
The paper focuses on user pairing and power allocation strategies in a mobile satellite environment. Authors have done a reasonably good work using DRL to group NOMA users. The document addresses the challenges of diverse delay quality-of-service (QoS) requirements in satellite networks and proposes a DRL-based solution for efficient user pairing. Simulations results show improved performance. However, authors shall mention the computational complexity and additional delay incurred due to DRL based user pairing. NOMA is already well known for higher end-to-end delay and additional DRL deployment will cause more delay. This is critical. The document does not explicitly mention the computational complexity or delay. While NOMA has a few advantages, it is important to note that there are also challenges and limitations associated with its implementation in satellite communications. These include channel estimation errors, co-channel interference, complexity, and mobility constraints. These must be mentioned in the document as well.
Why there are two TDMA with random selection strategy curves in Figure 6?
English language is not bad. However, don't use symbols as axis labels and in caption. Write the full word in English.
Author Response

(The authors gave the same response as above.)

Round 2
Reviewer 1 Report
The authors address most of my previous comments. However, some answers are not convinced. Especially, regarding the code-domain NOMA technique, which type of channel coding are popular? One should be LDPC codes, as stated in the references [A-C] in my previous-round comments. I believe the authors should provide a comprehensive survey on code domain NOMA and the associated LDPC codes by citing the related paper mentioned in the last round review. The should also clarify why they consider only power domain NOMA?
Is it possible to apply LDPC codes to improve the error performance of the considered NOMA systems.
Moreover, the computational complexity and time delay should be carefully analyzed so as to verify the proposed design.
Last but not the latest, some typos exist in the manuscript. Please make a careful proof reading.
All the above issuses should be carefully address before comfirming a final acceptance recommendation.
The authors address most of my previous comments. However, some answers are not convinced. Especially, regarding the code-domain NOMA technique, which type of channel coding are popular? One should be LDPC codes, as stated in the references [A-C] in my previous-round comments. I believe the authors should provide a comprehensive survey on code domain NOMA and the associated LDPC codes by citing the related paper mentioned in the last round review. The should also clarify why they consider only power domain NOMA?
Is it possible to apply LDPC codes to improve the error performance of the considered NOMA systems.
Moreover, the computational complexity and time delay should be carefully analyzed so as to verify the proposed design.
Last but not the latest, some typos exist in the manuscript. Please make a careful proof reading.
All the above issuses should be carefully address before comfirming a final acceptance recommendation.
Author Response
Attached please find our point-to-point response to reviewer 1 comments.

Reviewer 2 Report
The paper is improved. However, in the manuscript authors have not mentioned additional delay that might incur due to the deep learning approach. This must be done. No matter how many publications are there on NOMA and how superior NOMA is. This point is very important.
The English language usage is acceptable.
Author Response
Response to Reviewer 2 Comments
Comment1: In the manuscript authors have not mentioned additional delay that might incur due to the deep learning approach. This must be done. No matter how many publications are there on NOMA and how superior NOMA is. This point is very important.
Response: Thanks for your comment. To address your concern, we add a footnote in Section 4 on Page 8 in the revised manuscript “ Although DRL deployment will cause more delay, it is believed that this delay can be significantly decreased with the improvement of chip processing speed.”.